# SpecRA: Monitor Degenerative Repetition in LLM Agents using Randomized FFT

## Abstract

LLM-based agents also suffer from "degenerative repetition" like chatbots, which leads to task failure and results in significant waste of computational resources and API costs until token limit is reached. Existing methods require modification of training process or customization of model deployment, and detection algorithms are brittle to approximate or structural recurrence. We therefore introduce SpecRA, a simple yet effective algorithm for detection of self-repetitions in text. Via a randomized projection from the large LLM vocabulary onto a unit-norm complex sequence, our method leverages the power of the Fast Fourier Transform (FFT) to compute the sequence's autocorrelation. Peaks in the autocorrelation function robustly reveal the underlying periodicity of the content, with tolerance to minor variations. Through an analysis of 813 repetitive samples identified from 1.13M records of anonymized agent outputs, we build a taxonomy of repetition modes in agents and show that SpecRA offers a lightweight, non-intrusive mechanism for constructing more reliable and cost-efficient LLM agents across both standard open-source model deployments and proprietary models.

## 1 Introduction

The promise of LLM-based agents to automate complex, multi-step tasks is transforming diverse fields, from deep research and software engineering to scientific discovery and gaming (Huang et al., 2025b; Jimenez et al., 2024; Chen et al., 2025; Hu et al., 2024). However, their practical deployment is constrained by degenerative repetition, where the model becomes trapped in a recursive loop, generating near-identical sequences repeatedly (Holtzman et al., 2019; Huang et al., 2025a).

This behavior not only leads to task failure but also incurs significant computational waste or API costs. The severity is amplified by the scale of modern agent deployments: a single long-running agent task may involve dozens to hundreds of model invocations, meaning that even a seemingly low failure rate of 1 in 10,000 can affect a substantial number of tasks at scale. Moreover, these failures can cascade, as users may repeatedly re-trigger a malfunctioning agent, inadvertently creating a denial-of-service-like scenario that degrades performance for all concurrent users.

Current approaches prove largely inadequate for addressing this challenge. Penalty-based methods can mitigate repetition but often compromise overall performance (OpenAI, 2025; Keskar et al., 2019) and frequently require custom model deployments that are impractical for many applications (Dong et al., 2025; Ginart et al., 2025). Moreover, many commercial LLM APIs either fix decoding hyperparameters such as temperature or do not expose repetition-related penalties to end users, making it difficult for most agent developers to rely solely on decoding-time mitigation.

Classical exact string matching techniques (such as n-grams and suffix trees) fail entirely when faced with minor variations, while edit-distance algorithms suffer from prohibitive polynomial runtime complexity, making them unsuitable for streaming applications where pattern lengths are unknown a priori.

We introduce **SpecRA**, a fast spectral detector for approximate repetition. Rather than analyzing the text directly, we recast the input sequence as a discrete signal of constant energy and exploit tools from signal processing. Each token is projected to a uniformly distributed random complex of unit magnitude, producing a sequence $S = \{s_1, \ldots, s_N\}$. The projection makes non-repetitive text behave like white noise, so peaks provide a robust, quantitative signal for periodicity. We then

compute its autocorrelation of lag $k$ via the Wiener-Khinchin theorem:

$$R(k) = \mathcal{F}^{-1}\big(|\mathcal{F}(S)|^2\big)[k],$$

where $\mathcal{F}$ denotes the discrete Fourier transform. The procedure is streaming-friendly with overall $\mathcal{O}(N \log N)$ time.

Our contributions are: **(1)** Formulation of detection of degenerative repetition as approximate periodicity detection over discrete streams; **(2)** An efficient approximate autorepetition detection algorithm; **(3)** Theoretical bounds on false positives and detection efficacy; **(4)** A taxonomy of repetition modes from 1.13M agent traces and practical guidance on setting the detection threshold.

## 2 RELATED WORK

Degenerative repetition is a widespread issue affecting modern LLMs, including current frontier models. This phenomenon has been observed across diverse LLM-driven tasks such as code generation, translation, and dialogue (Dong et al., 2025; Wang et al., 2024; Xi et al., 2021). Prior research has analyzed its underlying causes and mechanisms, including exposure bias and likelihood-driven decoding that over-amplify frequent patterns, duplicated training data with skewed token frequencies, and high-inflow dynamics that trap the generation process in self-reinforcing attractors (Holtzman et al., 2019; Li et al., 2023; Fu et al., 2021; Mahaut & Franzon, 2025).

The most common approach involves applying penalties during the decoding process to discourage repetitive behavior. Frequency and presence penalties, popularized by OpenAI-compatible APIs, penalize tokens that have already appeared in the context window, while repetition penalty (Keskar et al., 2019) suppresses the generation of duplicate n-grams. However, their effectiveness is highly sensitive to hyperparameter tuning, and overly aggressive penalties can degrade output quality and coherence.

More advanced decoding techniques such as contrastive search (Su et al., 2022; Sen et al., 2025), information-theoretic penalties (Ginart et al., 2025), and grammar-aware penalties (Dong et al., 2025) have been proposed to further reduce repetition rates. Nevertheless, these methods remain less widely adopted, as they are not supported by many LLM inference providers or require custom model deployments. Alternative approaches such as model editing (Wang et al., 2024) target the problem at the model level but require significant computational effort and specialized expertise, making them impractical for most agent developers.

An alternative paradigm focuses on post-hoc detection rather than prevention during generation. Classical exact-match detection approaches include n-gram overlap and suffix trees, but these fail entirely when faced with minor lexical variations. While edit-distance methods (Landau et al., 1998) can tolerate some variations, they suffer from quadratic or higher complexity that becomes prohibitive for streaming applications processing long sequences. Specialized periodicity detection algorithms (Kolpakov & Kucherov, 1999; Main & Lorentz, 1984) achieve linear $\mathcal{O}(N)$ runtime but are designed specifically for exact repetitions and require computationally expensive extensions to handle approximate matching scenarios typical in LLM outputs.

Methods from bioinformatics offer alternative approaches to repetition detection (Kurtz et al., 2001). K-mer based techniques, widely used in genomic sequence analysis, are also brittle to minor variations. More promisingly, Fourier transforms have been successfully used in bioinformatics for detecting tandem repeats (Silverman & Linsker, 1986) by mapping nucleotides to complex symbols. SpecRA adapts this insight to LLM token vocabularies with randomized projection, achieving $\mathcal{O}(N \log N)$ complexity while maintaining robustness to lexical variation.

## 3 PROBLEM DEFINITION

**Task intuition.** Given a live token stream from an LLM agent, we want to raise an alarm as soon as the agent falls into a "loop", namely, when its output becomes approximately periodic after allowing up to $\varepsilon N$ mismatches per period.

While the excerpt below appears to show perfect repetition at first glance, the "P_P" sequence in the middle disrupts the otherwise regular pattern. This exemplifies *approximate periodicity*, which we formally define below.

**Excerpt from Gemini-2.5-Pro-0605 using temperature of 0.3**
*...normal output writing a markdown table...*
P_S_P_S_P_S_P_S_P_S_P_S_P_S_P_S_P_S_P_S_P_S_P_S_P_S_P_S_P_S_P_S_
P_S_P_S_P_S_P_S_P_S_P_S_P_S_P_S_P_S_P_P_S_P_S_P_S_P_S_P_S_P_S_P_
S_P_S_P_S_P_S_P_S_P_S_P_S_P_S_P_S_P_S_P_S_P_S_P_S_P_S_P_S_P_S_P_
S_P_S_P_S_P_S_P_S_P_S_P_S_P_S_P_S_P_S_P_S_P_S_P_S_P_S_P_S_P_

**Data model.** Let $V$ be a finite vocabulary and $\mathbf{x} = (x_1, x_2, \dots)$, $x_t \in V$, the potentially unbounded sequence of tokens emitted by an LLM agent. At time $t$ the first $t$ tokens are observable; future tokens are not.

**Approximate periodicity.** Fix an integer period length $p \geq 2$ and an error budget $\varepsilon \in [0, 1)$. Given an index $s$ and positive integer $K$, denote by

$$B_j(s,p) \; = \; (x_{s+(j-1)p+1}, \dots, x_{s+jp}), \quad j = 1, \dots, K,$$

the $j$-th contiguous block of length $p$. We say the window $\mathbf{x}_{s:s+Kp-1}$ is $(\varepsilon, p)$-*approximately $K$-periodic* if there exists a reference block $U \in V^p$ such that

$$\frac{1}{p} d\big(B_j(s,p), U\big) \; \leq \; \varepsilon, \quad \text{for every } j = 1, \dots, K,$$

where $d(\cdot, \cdot)$ is a token-level distance (e.g. Hamming distance or edit distance). In words, each block differs from the reference pattern in at most an $\varepsilon$ fraction of its positions.

Remark: For theoretical clarity, we assume fixed-length period blocks in our analysis. Extensions to handle variable-length blocks due to insertions and deletions are discussed in Section 7.

**Definition 1** (Degenerative-repetition event). A stream $\mathbf{x}$ *enters degenerative repetition* at time $t_0$ if there exist $p \in [P_{\min}, P_{\max}]$, $K \geq K_{\min}$, and $\varepsilon \leq \varepsilon_{\max}$ such that $\mathbf{x}_{t_0:t_0+Kp-1}$ is $(\varepsilon, p)$-approximately $K$-periodic according to the criteria above.

**Online detection task.** At each time $t$ the detector outputs a Boolean alarm $A_t \in \{0, 1\}$. A correct detector should satisfy two properties for given false-alarm probability $\delta$ and detection delay $D$:

  (i) (**Low false positives**) For any stream that never satisfies Definition 1, $\Pr[A_t = 1] \leq \delta$ for all $t$.

  (ii) (**Timely detection**) If a degenerative repetition event starts at $t_0$, then with probability at least $1 - \delta$ the detector raises an alarm no later than $t_0 + D$, i.e. $\exists t \leq t_0 + D$ with $A_t = 1$.

**Streaming constraints.** We adopt the standard RAM streaming model:

  • **Per-token time** must be sub-linear in the window size; our target is $\mathcal{O}(\log W)$ amortized per token, achieved via FFT.
  • **Memory** is $\mathcal{O}(W)$, where $W$ is the largest sliding-window length the detector inspects.

**Objective.** Design an algorithm that, for user-specified $(\varepsilon_{\max}, P_{\min}, P_{\max}, K_{\min}, \delta, D, W)$, meets the guarantees above while respecting the streaming constraints.

The subsequent sections show that SpecRA meets these requirements with $\mathcal{O}(W \log W)$ preprocessing per window and $\mathcal{O}(\log W)$ time per arriving token.

## 4 METHODOLOGY

**Overview.** SpecRA transforms the discrete token detection problem into a continuous signal processing task through three key stages: (i) **randomized projection** maps each token to a unit-magnitude complex number, converting the discrete vocabulary into a continuous signal while preserving repetition structure; (ii) **spectral analysis** computes the autocorrelation function via

FFT, efficiently identifying periodic patterns across multiple candidate periods; and (iii) **statistical detection** compares the maximum autocorrelation peak against a threshold derived from theoretical false-positive bounds.

The core insight is that repetitive text exhibits strong autocorrelation peaks at the repetition period, while non-repetitive text behaves like white noise with near-zero autocorrelation. The randomized projection ensures robustness to minor variations (e.g., number increments, minor spelling changes) that would confound exact string matching approaches.

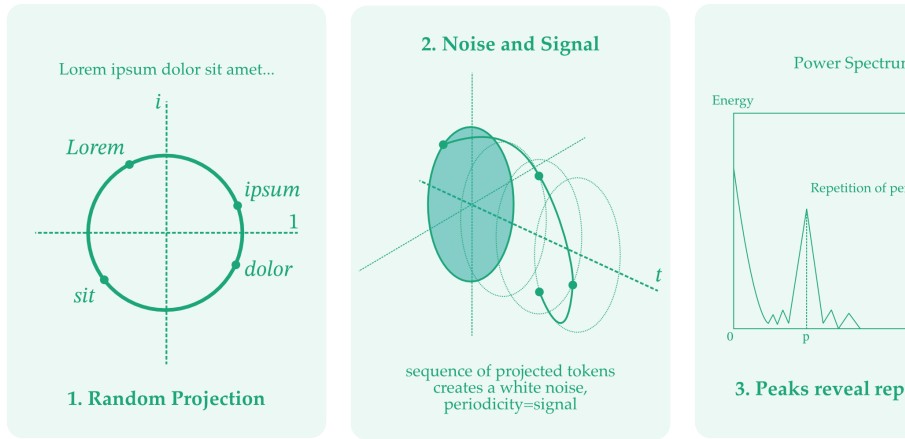

Figure 1: SpecRA workflow diagram showing the complete pipeline from token stream input to periodicity detection output.

**Randomized Token Projection.** Let $V$ be the model's vocabulary of size $|V|$. For each token $v \in V$, we draw an independent random phase $\theta_v \sim \mathcal{U}[0, 2\pi)$ and define the projection function $\phi : v \mapsto e^{i\theta_v}$. This maps each token to a point on the unit circle in the complex plane, ensuring constant signal energy $|\phi(v)| = 1$ regardless of token identity. This constraint is crucial: if tokens had unequal magnitudes, those with larger norms would disproportionately dominate the autocorrelation, biasing the repetition score based on token magnitude rather than structural periodicity. By enforcing unit magnitude, we treat all tokens symmetrically.

Given a token stream $(x_1, x_2, \dots)$, we obtain the complex sequence $S = (s_1, s_2, \dots)$ where $s_t = \phi(x_t)$. The key property is that identical tokens always map to the same complex number, preserving exact repetition structure, while different tokens map to (nearly) orthogonal directions. This design makes the detector robust to lexical variations: swapping "large" with "big" changes the complex representation but preserves the overall periodic structure if the substitution occurs consistently across repetitions.

**Spectral Autocorrelation.** We apply the Wiener-Khinchin theorem to compute the autocorrelation efficiently via FFT. For a sliding window of length $W$, the circular autocorrelation at lag $k$ is:

$$R_k = \mathcal{F}^{-1}\big(|\mathcal{F}(S)|^2\big)[k] = \sum_{t=1}^{W} s_t s_{t-k}^*,$$

where $\mathcal{F}$ denotes the discrete Fourier transform and $s_{t-k}^*$ is the complex conjugate with indices taken modulo $W$.

The power spectrum $|\mathcal{F}(S)|^2$ captures the frequency content of the signal, and its inverse FFT yields the autocorrelation across all lags simultaneously. For repetitive sequences with period $P$, the autocorrelation $R_P$ exhibits a large magnitude because many terms $s_t s_{t-P}^*$ align constructively. For non-repetitive sequences, these terms behave like independent random rotations, resulting in near-zero autocorrelation due to destructive interference.

**Repetition Score.** To detect repetitive patterns, we focus on the real part of the autocorrelation, which captures the alignment between tokens at different lags. Let $P_{\min}$ and $P_{\max}$ define the range

of plausible periods (e.g., 2 to 256 tokens). We compute the normalized repetition score:

$$S_{\text{rep}} = \max_{l=P_{\min}}^{P_{\max}} \frac{\Re(R_l)}{\Re(R_0)}$$

The denominator $\Re(R_0) = W$ normalizes by the total signal energy, ensuring the score is invariant to window size. The numerator $\Re(R_l) = \sum_{t=1}^{W} \Re(s_t s_{t-l}^*) = \sum_{t=1}^{W} \cos(\theta_t - \theta_{t-l})$ measures how well the sequence aligns with itself in the sense of cosine similarity when shifted by $l$ positions. For perfect repetition with period $P$, we have $S_{\text{rep}} \approx 1$, while for random sequences, $S_{\text{rep}} \approx 0$.

We trigger a repetition alarm when $S_{\text{rep}} > S_{\text{th}}$ for some threshold $S_{\text{th}} \in (0,1)$. The period range $[P_{\min}, P_{\max}]$ excludes trivial cases: periods smaller than $P_{\min} = 2$ are not meaningful, while periods larger than $P_{\max}$ would require prohibitively long sequences to establish reliable patterns.

---

**Algorithm 1:** SPECRA-BATCH: Batch processing with FFT

---

**Input:** token stream $(x_1, x_2, \dots)$, window size $W$, period range $[P_{\min}, P_{\max}]$, threshold $S_{\text{th}}$,
     batch size $B$ (e.g., $B = W$)
**Output:** alarm bit $A_t$ every $B$ timesteps
**Offline:** draw phases $\theta_v \sim \mathcal{U}[0, 2\pi]$ and set $\phi(v) = e^{i\theta_v}$;
**Initialize:** batch buffer $\mathcal{B}$ of size $B$;
**Online:**
**for** $t \leftarrow 1, 2, \dots$ **do**
    $s_t \leftarrow \phi(x_t)$;
    append $s_t$ to $\mathcal{B}$;
    **if** $|\mathcal{B}| = B$ **or** *end-of-stream* **then**
        $F \leftarrow \mathcal{F}(\mathcal{B})$;
        $R \leftarrow \mathcal{F}^{-1}(|F|^2)$;
        $S_{\text{rep}} \leftarrow \max_{l=P_{\min}}^{P_{\max}} \frac{\Re(R_l)}{\Re(R_0)}$;
        $A_{\text{batch}} \leftarrow \mathbf{1}\{S_{\text{rep}} > S_{\text{th}}\}$;
        emit $A_{\text{batch}}$;
        clear $\mathcal{B}$;
    **end**
**end**

---

**Computational Complexity.** Initializing one window costs $\mathcal{O}(W \log W)$. Afterwards each batch has a complexity of $\mathcal{O}(W \log W)$ and can be amortized to $\mathcal{O}(\log W)$ per token when processed in batches, meeting the streaming constraints of Section 3.

## 5    EFFECTIVENESS ANALYSIS

### 5.1    BEHAVIOR UNDER THE NULL HYPOTHESIS

Under a non-repetitive stream, the projected tokens $s_t = e^{i\theta_t}$, with $\theta_t \sim \mathcal{U}[0, 2\pi]$, form i.i.d. isotropic noise. For any non-zero lag $l \neq 0$, the real part of the circular autocorrelation, $\Re(R_l) = \sum_{t=1}^{W} \Re(s_t s_{t-l}^*)$, is a sum of $W$ i.i.d. random variables. Each term $\Re(s_t s_{t-l}^*) = \cos(\theta_t - \theta_{t-l})$ is a random variable bounded in $[-1, 1]$ with zero mean.

By applying Hoeffding's inequality to this sum, we can bound the probability of observing a large repetition score purely by chance. Let $M = P_{\max} - P_{\min} + 1$ be the number of candidate periods. A union bound over these periods yields:

**Lemma 1** (False-positive bound). *For any threshold $S_{th} > 0$,* $\Pr_{\text{null}}[S_{rep} > S_{th}] \leq M \cdot \exp\left(-\frac{W S_{th}^2}{2}\right)$.

Choosing $S_{\text{th}*} = \sqrt{\frac{2}{W} \log(M/\delta)}$ guarantees a false-alarm rate no greater than $\delta$.

*Proof.* Under the null hypothesis, the projected tokens $s_t = e^{i\theta_t}$ are i.i.d. with phases $\theta_t$ uniformly distributed in $[0, 2\pi)$. For any non-zero lag $l$, the term $s_t s_{t-l}^* = e^{i(\theta_t - \theta_{t-l})}$ is a random rotation. Let $Y_t = \Re(s_t s_{t-l}^*) = \cos(\theta_t - \theta_{t-l})$. Since $\theta_t$ and $\theta_{t-l}$ are independent and uniformly distributed, their difference modulo $2\pi$ is also uniform in $[0, 2\pi)$.

The variables $\{Y_t\}_{t=1}^W$ are thus i.i.d.[1] with $\mathbb{E}[Y_t] = \frac{1}{2\pi} \int_0^{2\pi} \cos(u) du = 0$ and are bounded in the interval $[-1, 1]$.

The real part of the autocorrelation is $\Re(R_l) = \sum_{t=1}^W Y_t$. The repetition score for this lag is $\frac{\Re(R_l)}{R_0} = \frac{1}{W} \sum_{t=1}^W Y_t$, since $R_0 = W$. We want to bound the probability of the event $\{\frac{\Re(R_l)}{W} > S_{\text{th}}\}$, which is equivalent to $\{\sum_{t=1}^W Y_t > W S_{\text{th}}\}$.

We apply Hoeffding's inequality. For a sum $S_W = \sum Y_t$ of $W$ independent random variables where $Y_t \in [a_t, b_t]$, the inequality states $\Pr[S_W - \mathbb{E}[S_W] \geq \epsilon] \leq \exp(-\frac{2\epsilon^2}{\sum (b_t - a_t)^2})$. Here, $\mathbb{E}[S_W] = 0$, $\epsilon = W S_{\text{th}}$, $a_t = -1$, and $b_t = 1$, so $b_t - a_t = 2$. For a single lag, we have:

$$\Pr\left[\frac{\Re(R_l)}{W} > S_{\text{th}}\right] \leq \exp\left(-\frac{2(W S_{\text{th}})^2}{\sum_{t=1}^W (1 - (-1))^2}\right) = \exp\left(-\frac{2W^2 S_{\text{th}}^2}{4W}\right) = \exp\left(-\frac{W S_{\text{th}}^2}{2}\right).$$

The score $S_{\text{rep}}$ is the maximum over $M = P_{\max} - P_{\min} + 1$ candidate lags. Applying the union bound gives the final result:

$$\Pr_{\text{null}}\left[S_{\text{rep}} > S_{\text{th}}\right] \leq \sum_{l=P_{\min}}^{P_{\max}} \Pr\left[\frac{\Re(R_l)}{W} > S_{\text{th}}\right] \leq M \cdot \exp\left(-\frac{W S_{\text{th}}^2}{2}\right),$$

proving the claim. $\qquad\square$

## 5.2 POWER UNDER $\varepsilon$-MISMATCH APPROXIMATE PERIODICITY

Assume a true period $P$. For each position $t$, with probability $1 - \varepsilon$ we have an exact repeat $x_t = x_{t-P}$; with probability $\varepsilon$ a mismatch occurs where $x_t$ is independent of $x_{t-P}$ (and independent across $t$). Under the fixed random projection $\phi$ above, define $X_t = \Re(s_t s_{t-P}^*) \in [-1, 1]$.

When $x_t = x_{t-P}$, $X_t = 1$; when a mismatch occurs, $s_t$ and $s_{t-P}$ are independent unit phases so $\mathbb{E}[X_t] = 0$. Therefore

$$\mathbb{E}[X_t] = (1 - \varepsilon) \cdot 1 + \varepsilon \cdot 0 = 1 - \varepsilon, \quad \mathbb{E}[\Re(R_P)] = \sum_{t=1}^W \mathbb{E}[X_t] = W(1 - \varepsilon).$$

Since $R_0 = W$, the normalized score for the true period is $\frac{\Re(R_P)}{R_0} = \frac{1}{W} \sum_{t=1}^W X_t$ with mean $1 - \varepsilon$.

**Theorem 1** (Exponential bound under $\varepsilon$-mismatch). *If $0 < S_{th} < 1 - \varepsilon$ and mismatches occur independently across $t$, then*

$$\Pr[S_{rep} \leq S_{th}] \leq \exp\left(-\frac{W(1 - \varepsilon - S_{th})^2}{2}\right).$$

*Proof.* Set $S_W = \sum_{t=1}^W X_t$ and $\mu' = \mathbb{E}[S_W] = W(1 - \varepsilon)$. Each $X_t \in [-1, 1]$ and, by assumption, the $\{X_t\}$ are independent. The miss event $\{S_{\text{rep}} \leq S_{\text{th}}\}$ implies $\Re(R_P) \leq S_{\text{th}} W$, i.e., $S_W - \mu' \leq -(\mu' - S_{\text{th}} W)$. Hoeffding's inequality yields

$$\Pr[S_W - \mu' \leq -(\mu' - S_{\text{th}} W)] \leq \exp\left(-\frac{(\mu' - S_{\text{th}} W)^2}{2W}\right) = \exp\left(-\frac{W(1 - \varepsilon - S_{\text{th}})^2}{2}\right).$$

$\qquad\square$

---

[1] Although terms like $Y_t$ and $Y_{t+l}$ share the phase $\theta_t$, the phase differences $(\theta_t - \theta_{t-l}) \pmod{2\pi}$ remain pairwise independent. The only statistical dependency in the full sum arises from the circular boundary condition. To ensure strict validity for Hoeffding's inequality, we may discard the $l$ wrap-around terms, which breaks the circular dependency and makes the remaining terms strictly i.i.d., with negligible energy loss since $W \gg l$. For clarity we write $W$ as the window length without the wrap-around terms.

# 6 EMPIRICAL ANALYSIS

## 6.1 ROBUSTNESS AGAINST SYNTHETIC NOISE

Theoretical analysis in Section 5 suggests that SpecRA can resist minor substitutions, while the effect of different noise levels and robustness against insertions and deletions remain to be investigated. We empirically validate this by generating synthetic sequences and evaluating the repetition scores.

To isolate the effect of perturbations we generate synthetic sequences in the form $T = \underbrace{(P \parallel P \parallel \ldots \parallel P)}_{L/p \text{ copies}} \oplus \mathcal{N}(\varepsilon)$, where $P$ is a base pattern of length $p$ drawn uniformly at random from a vocabulary of size $V = 32768$, $L = 1024$ is the total window length, and $\mathcal{N}(\varepsilon)$ applies one of substitution, deletion and insertion at rate $\varepsilon \in [0, 0.2]$. We tested $p \in \{4, 16, 64\}$ and report the median repetition score $S_{\text{rep}}$ over $10^4$ Monte-Carlo trials per setting.

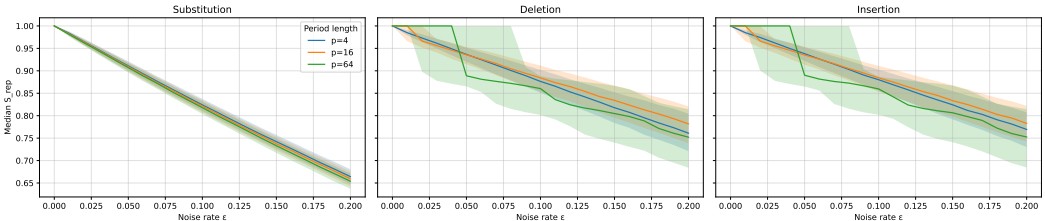

Figure 2: Median repetition score $S_{\text{rep}}$ as a function of noise rate $\varepsilon$. Shaded bands denote the inter-quartile range. Insertions and deletions apply to the same randomly chosen anchor index within each period copy to simulate structural repetitions that LLMs produce.

For substitution noise, curves for different $p$ almost overlap and $S_{\text{rep}}$ decays almost linearly with $\varepsilon$ for all $p$, confirming its insensitivity to noise level and the underlying period length under substitutions. SpecRA also tolerates indels on short patterns, as long as major structure is preserved.

To better approximate the token statistics of natural language, we further evaluate repetition scores on synthetic sequences where both the base pattern $P$ and the corruption noise are drawn from a truncated Zipf distribution over the vocabulary.

In this setting we treat edit distance between adjacent period copies as a gold standard measure of autorepetition, and compare SpecRA against several practical heuristics. As pairwise sequence comparison methods, SimHash and MinHash are not directly applicable, so they are given access to the true period $P$ (Oracle). NaiveFFT directly translates classical FFT-based approximate string matching by mapping $n$-th token to corresponding unit root $n \mapsto e^{2\pi i n/V}$. Experiment details are provided in Appendix D.

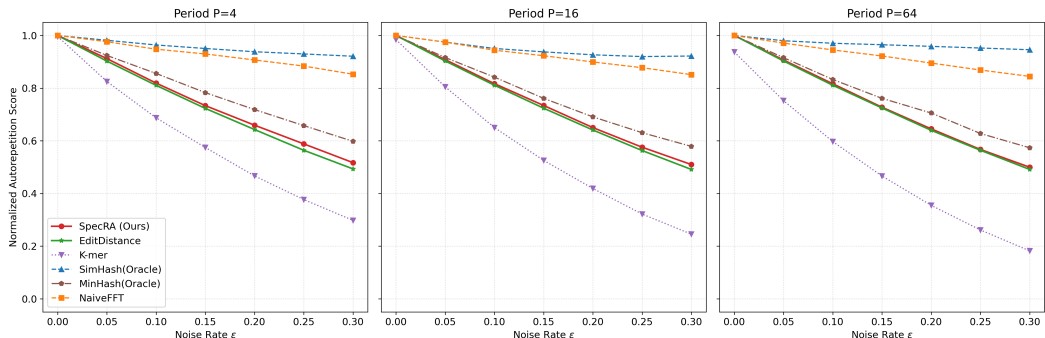

Figure 3: Normalized autorepetition score as a function of substitution noise rate $\varepsilon$ on synthetic sequences with Zipfian token frequencies. The closer to EditDistance, the better.

Across all noise levels and period lengths, SpecRA almost exactly matches the EditDistance curve, indicating that our spectral self-similarity score is a highly calibrated proxy for edit distance.

In contrast, NaiveFFT and SimHash systematically overestimate repetition and fail to decay appropriately with noise, maintaining high scores even at high noise levels. On the other hand, the K-mer heuristic underestimates repetition and decays too aggressively. While MinHash follows the decay trend better, it still lacks the precision of our method. SpecRA thus achieves a near-oracle approximation to edit distance while being the fastest method among all baselines (Fig. 4).

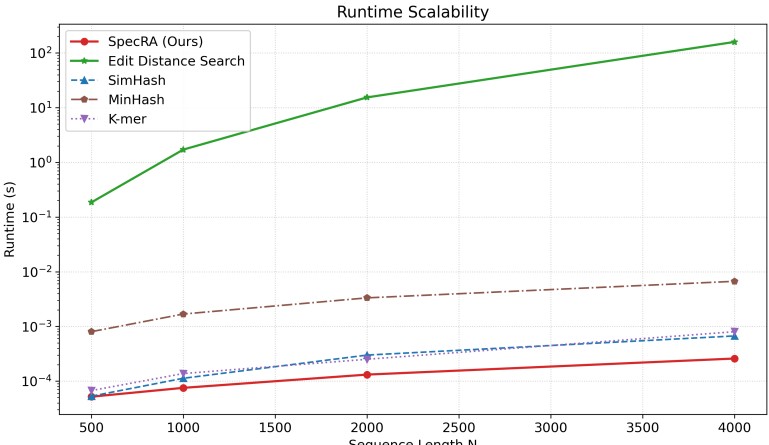

Figure 4: Runtime performance comparison between SpecRA and other baselines.

## 6.2 REPETITION FEATURES IN REAL-WORLD DATA

To properly set the detection threshold for SpecRA, we need to first understand the distribution of the repetition score $S_{\text{rep}}$ in real-world data. We sampled 153,060 passages from Wikipedia (89,359 passages in English and 63,701 passages in Chinese) (Foundation, 2023), 208,414 code snippets from GitHub Code (CodeParrot, 2022), and collected repetition scores from 1,133,797 rounds of LLM outputs from a general-purpose agent powered by state-of-the-art proprietary models (or their equivalents with >100B parameters) working on tasks spanning different domains involving software development, data processing, analysis/visualization, and general NLP. The $S_{\text{rep}}$ distribution is shown in Figure 5.

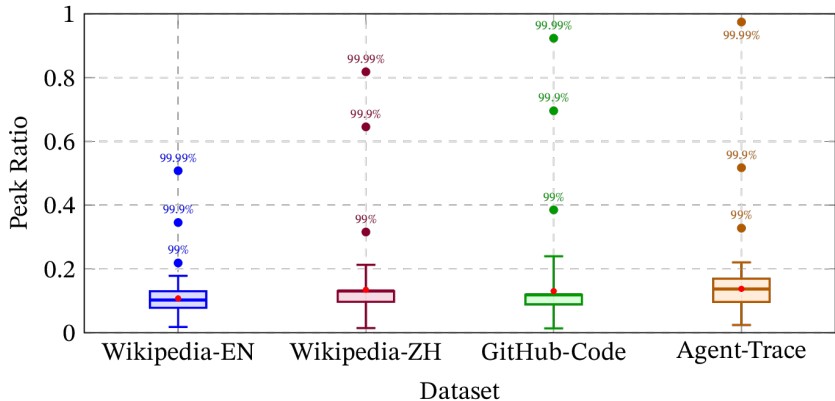

Figure 5: Repetition score distribution comparison across Wikipedia-EN, GitHub-Code, and Agent-Trace datasets. Red dots indicate mean values.

Industrial deployments that run the detector on all text streams (e.g., logging, analytics) typically demand far stricter guarantees: FPR $< 10^{-3}$ or even $< 10^{-4}$. Empirically the GitHub-Code corpus dominates the upper tail, with the 99.9th and 99.99th percentiles located at $S_{99.9} = 0.69$ and $S_{99.99} = 0.92$, respectively. We therefore recommend the following tiers:

- **Balanced:** $S_{\mathrm{th}} = 0.69$   for general applications
- **Safe:** $S_{\mathrm{th}} = 0.92$   for coding agents

Users with custom vocabularies or window sizes can re-estimate the quantile with a single offline scan and plug them into the same decision rule.

## 6.3 AGENT REPETITION TAXONOMY

SpecRA flagged 813 suspicious repetition samples out of 1,133,797 agent traces (0.071%), using a threshold of $S_{\mathrm{th}} = 0.69$. We excluded 264 samples that were too short to classify reliably, likely due to incomplete generation from network errors or early termination. The remaining 549 samples were classified into four distinct repetition categories (see Appendix E for annotation details):

**Structural repetition.**   Systematic iteration over semantically related content patterns, where agents generate sequences of structurally similar elements with incremental variation. Examples include enumeration of chemical elements in periodic table order, systematic generation of numbered function definitions, and iterative construction of similar data structures. This pattern reflects the model's attempt to complete structured tasks through template-based generation.

**Syntactic degradation.**   Purely syntactic repetition without semantic coherence, where models generate identical token sequences or character patterns with no underlying logical structure. This includes infinite repetition of single characters, alphabetical and numerical sequences (e.g., ",3,3,3..."), representing complete semantic breakdown in the generation process.

**Binary data generation.**   A notable repetition pattern observed in agents attempting to emit binary data (105 / 549; **19.1%**), manifesting in three distinct sub-patterns: **(1) Multimedia encoding loops:** cyclical repetition of base64-encoded character sequences representing multimedia content (images, audio, video files) or structured documents; **(2) URI malformation cycles:** iterative generation of malformed data URIs or embedded content for visualization services (e.g., mermaid.ink, plantuml.com), often producing corrupted markup with repeated URI fragments; and **(3) Direct binary emission:** direct attempts to emit binary file headers and control characters (e.g., ZIP/Office file magic numbers "PK", PNG signatures), interspersed with repeated \uFFFE patterns.

**Legitimate repetition.**   Cases where structurally necessary repetition is misclassified as degenerative, including large-scale data serialization (JSON arrays, CSV records), ASCII art containing repeated patterns, systematic progress tracking with templated status reports, and algorithmic output requiring repetitive formatting patterns that serve a functional purpose.

These categories constitute 46.26%, 21.68%, 19.13%, 12.93% of the flagged cases, respectively.

Table 1: Breakdown of the 549 repetitive turns by error category ($S_{\mathrm{th}} = 0.69$).

| Category | # samples | Share |
|---|---:|---:|
| Structural repetition | 254 | 46.26% |
| Syntactic degradation | 119 | 21.68% |
| Binary total | 105 | 19.13% |
| Multimedia encoding loops | 55 | 10.02% |
| URI malformation cycles | 31 | 5.65% |
| Binary header emission | 19 | 3.47% |
| Legitimate repetition | 71 | 12.93% |
| **Total** | **549** | 100% |

## 7 DISCUSSION

### 7.1 USAGE NOTE ON PRACTICAL DEPLOYMENT

For agent developers who rely on LLM API providers without access to inference infrastructure, original tokens may not be available. In such cases, SpecRA can still be effectively applied to character-level streams to detect repetitive failures. However, when applied to smaller vocabularies (e.g., ASCII-only streams), performance may degrade due to increased collision probability in the hash space.

Phase values $\theta_v$ are sampled i.i.d. from a continuous distribution. A potential issue arises when two distinct tokens are assigned similar phases, causing mismatches to contribute $\cos(\Delta) \approx 1$ to the autocorrelation, mimicking matches.

The risk depends on the interplay between random phase collisions and token co-occurrence statistics. Large vocabularies increase the number of potential collision pairs ($\binom{|V|}{2}$), while small vocabularies concentrate statistical weight on fewer pairs. If frequently co-occurring tokens happen to receive similar phases, the distortion effect is amplified.

Our proposed mitigation using $K$ independent projections is highly effective. For a mismatch to consistently distort the signal, it must be a near-collision across all $K$ mappings—a vanishingly unlikely event that ensures detector reliability regardless of vocabulary size or input statistics.

Additionally, legitimate repetition may occasionally be misclassified as degenerate. In such cases, LLMs can serve as a secondary validation mechanism.

### 7.2 LIMITATIONS

While SpecRA demonstrates effectiveness in detecting repetitive failures, it has several inherent limitations. First, although SpecRA excels at identifying simple structural repetitions, it may struggle with more complex patterns that require deeper contextual understanding. For instance, it may fail to detect repetitive failures in code generation tasks that produce semantically similar code snippets with varying lengths, as the length variations introduce phase shifts across repetitive blocks (Dong et al., 2025).

Within the $(\varepsilon, p)$-approximate periodicity framework of Section 3, we did not observe false negatives on our agent-trace dataset, consistent with the exponential miss-detection bound in Theorem 1. However, more semantic forms of repetition that violate the fixed-period assumption, such as grammatically repetitive code blocks whose lengths grow across iterations, fall outside this formal setting and remain an important direction for future extensions of SpecRA.

As discussed in Section 7.1, SpecRA's performance is sensitive to vocabulary size. Smaller vocabularies increase collision probability, potentially leading to performance degradation. While we have proposed mitigation strategies using multiple independent projections, their effectiveness requires further empirical evaluation.

## 8 CONCLUSION AND FUTURE WORK

We framed degenerative repetition in LLM agents as an approximate periodicity detection problem and introduced **SpecRA**, which combines randomized phase projection with FFT-based autocorrelation analysis. Our method achieves $\mathcal{O}(W \log W)$ processing complexity with $\mathcal{O}(\log W)$ amortized time per token, while providing provable bounds on both false-alarm and miss-detection probabilities. Extensive experiments across public corpora and real agent traces demonstrate that SpecRA offers a lightweight, non-intrusive solution for building more reliable and cost-efficient LLM agents.

Future work can extend this research in three immediate avenues: **(1) Inference-time integration**, by incorporating SpecRA scores as decoding penalties to steer models away from repetitive attractors; **(2) Cross-modal generalization**, by adapting the spectral approach to detect cyclic artifacts in vocoder waveforms, embedding streams, or tool-use trajectories; and **(3) An enhanced signal-processing toolkit**, exploring techniques like wavelet coherence or adaptive filtering to build a comprehensive suite of guards for trustworthy AI.

## REPRODUCIBILITY STATEMENT

To ensure reproducibility, we provide a reference Python implementation in Appendix C. Our experiments use synthetic data, generated as described in Section 6.1, and public corpora (Wikipedia, GitHub Code) detailed in Section 6.2. The full experimental setup and parameters are specified in Section 6.

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

## A  PRIVACY AND ETHICS CONSIDERATIONS

Our analysis of agent logs was conducted under strict privacy safeguards and ethical guidelines. All LLM outputs were anonymized and de-identified prior to analysis, with access restricted exclusively to patterns flagged as anomalous by our detection algorithm. No personally identifiable information, proprietary content, or sensitive user-generated data was examined during the analysis process.

SpecRA provides inherent privacy advantages: it operates on statistical properties of token sequences rather than semantic content, enabling detection of repetitive failures without requiring persistent storage or detailed inspection of user data. This design preserves the confidentiality of user-agent interactions while delivering essential protection against computational waste and system instability. Furthermore, the randomized projection mechanism ensures that even if projection parameters were compromised, recovering original token sequences would remain impractical.

## B  USE OF LLMs

LLMs were utilized during various stages of this paper's development. Specifically, LLMs assisted with: (a) translating our original derivations and algorithmic ideas into initial LaTeX formulations and suggesting more standard notation, (b) language polishing and stylistic refinement, and (c) comprehensive proofreading. The core ideas, problem formulation, and all mathematical proofs, including Lemma 1 and Theorem 1, were developed and verified by the authors, who take full responsibility for their validity. We acknowledge that [Anonymous frontier model] helped us identify a flaw in an early draft of the proof of Theorem 1, we subsequently corrected and validated the final version ourselves.

Additionally, LLMs were employed to assist with the initial classification and anonymization of potentially suspicious repetitive cases prior to manual analysis. All automated labels were subsequently verified and when necessary, corrected by the authors to ensure accuracy and consistency.

## C  REFERENCE IMPLEMENTATION

This section provides a minimal Python implementation of the SpecRA algorithm. The code demonstrates the core spectral analysis pipeline described in Section 4, implementing offline batch processing suitable for research and prototyping.

```python
import numpy as np
from typing import List, Tuple

# Build a random unit-magnitude complex phase mapping for each token id.
# In practice, VOCAB_SIZE should be at least the vocabulary size of
# all models being used.
VOCAB_SIZE = 32768
SPECRA_MAP = np.exp(1j * 2 * np.pi * np.random.rand(VOCAB_SIZE))

# SpecRA detects approximate repetition in a token sequence
# using spectral analysis.
# Parameters:
#   - token_ids: input token sequence as integer ids
#   - threshold: detection threshold for normalized repetition score, range [0, 1]
# Returns:
#   - peak_ratio: maximum normalized autocorrelation value
#   - detected: True if repetition detected above threshold
def specra(token_ids: List[int], threshold: float) -> Tuple[float, bool]:
    # Step 1: Random projection
    seq = np.array([SPECRA_MAP[t] for t in token_ids], dtype=np.complex128)
    n = len(seq)

    # Step 2: FFT-based autocorrelation via Wiener-Khinchin theorem
    # Forward FFT: F = FFT(seq)
    coeffs = np.fft.fft(seq)
    # Power spectrum: power = |F|^2
```

```
power = np.abs(coeffs) ** 2
# Inverse FFT: autocorr = IFFT(|F|^2)
autocorr = np.fft.ifft(power)

# Step 3: Compute normalized repetition score
r0 = float(np.real(autocorr[0]))
if r0 <= 0.0:
    return 0.0, False
p_min, p_max = 1, n // 2 + 1
if p_max <= p_min:
    return 0.0, False

real_corr = np.real(autocorr[p_min:p_max])
peak = float(np.max(real_corr)) if real_corr.size > 0 else 0.0
return peak / r0, peak > threshold * r0
```

## D  SYNTHETIC NOISE EXPERIMENT DETAILS

**Data Generation.**  We generate synthetic repetitive sequences of fixed window length $W = 1{,}024$ with underlying periods $p \in \{4, 16, 64\}$ and vocabulary size of $V = 32{,}768$.

For the Uniform Noise setting, base patterns are drawn uniformly from the vocabulary, and noise replaces tokens with new ones drawn uniformly at random. For the Zipfian Noise setting, to better approximate natural language statistics, tokens are drawn from a Zipf distribution with exponent $a = 1.05$. Substitutions also draw replacement tokens from the same distribution, ensuring the noise follows the underlying term frequency distribution and maintaining the "long tail" property of language. For Indel Noise, we apply insertions and deletions at $\varepsilon \cdot N$ randomly chosen positions, fixing the mutation positions across all periods.

**Baselines Implementation.**  All baselines are implemented in Python. Scores are normalized to $[0, 1]$, where 1 corresponds to perfect repetition and 0 indicates that every token is unique. We compare SpecRA against:

- **EditDistance:** We compute the average edit distance between adjacent non-overlapping blocks of length $p$ in the sequence. We report $1 - (\text{distance}/p)$ as the similarity score. This baseline represents an idealized detector that knows the period $p$ but not the ground truth pattern.

- **EditDistance Search:** We search for the period $p$ in the sequence using a brute-force approach with period range $[2, N/10]$.

- **NaiveFFT:** A direct spectral method that maps token ID $k$ to the $k$-th root of unity $e^{2\pi i k / V}$ before applying FFT. This tests whether randomized projection is necessary compared to deterministic mapping.

- **K-mer:** We use $K = 4$. The score is calculated as $1 - \frac{\text{unique } K\text{-mers}}{\text{total } K\text{-mers}}$, measuring the redundancy of local sub-patterns.

- **SimHash:** We compute 64-bit SimHash fingerprints for each period-sized block. In practice, SimHash is usually used together with stop-word removal or TF-IDF, while in a streaming setting, TF-IDF is not feasible. For Zipfian data, we remove the top-10 most frequent tokens as stop-words to improve robustness, as high-frequency tokens dominate the hash signatures. We compare these fingerprints against the oracle pattern's fingerprint, giving this baseline a significant advantage.

- **MinHash:** We generate signatures using 16 permutations and compute Jaccard similarity against the oracle pattern's signature.

Runtime benchmarks were conducted on a standard CPU (Apple M1). We measured the average wall-clock time over 10 trials for sequence lengths $N \in \{500, 1000, 2000, 4000\}$. SpecRA's complexity is dominated by the FFT operation ($\mathcal{O}(N \log N)$), whereas edit distance search scales poorly ($\mathcal{O}(N^3)$).

# E   AGENT REPETITION TAXONOMY DETAILS

The 549 repetitive samples were annotated by multiple LLMs with major voting, and then carefully verified and when necessary, corrected by the authors.

The 264 examples were excluded because they were too short or incomplete to be classified reliably (e.g., due to network failure, user cancellation or safety filters). While they manifest repetitive features, it is uncertain whether the model would have entered an infinite loop (or equivalently whether the model will exit eventually after a some more periods). This may slightly shift the distribution, but likely not significantly.

During annotation, we allowed for an "Other" label for edge cases that are hard to classify, and manually labeled them after thorough inspection. The process of classification can be described as follows:

(i) Does the pattern carry coherent meaning to a human reader?
- Yes → (ii)
- No → (iii)
- Uncertain → (Other)

(ii) Given that the content is meaningful, is the repetition legitimate for the user's task (e.g., translating a repetitive table)?
- Yes → (Legitimate repetition)
- No → (Structural repetition)
- Uncertain → (Other)

(iii) Given that the content is not meaningful to humans, does it still correspond to machine-interpretable binary formats (e.g., base64-encoded multimedia content, binary file headers)?
- Yes → (Binary data generation)
- No → (Syntactic degradation)
- Uncertain → (Other)

