# OpenReview forum: "SpecRA: Monitor Degenerative Repetition in LLM Agents using Randomized FFT"
_ICLR.cc/2026/Conference — Submitted to ICLR 2026_

### Official Review · Reviewer_8wU1 · 2025-10-17

**Soundness:** 3
**Presentation:** 2
**Contribution:** 2
**Rating:** 4
**Confidence:** 2

**Summary:**

The paper addresses the issue of degenerative repetition in large language model (LLM)-based agents, where models get stuck in repetitive cycles, leading to wasted computational resources and increased API costs. The authors propose SpecRA, a fast, spectral method for detecting self-repetitions in LLM outputs. By leveraging the Fast Fourier Transform (FFT) and a randomized token projection, the method analyzes the periodicity of token sequences and identifies when repetition occurs. The paper presents a detailed taxonomy of repetition modes and demonstrates that SpecRA can reliably detect and classify repetitions across a range of LLM outputs. The authors argue that their method provides an efficient, non-intrusive mechanism to reduce computational waste and improve agent reliability.

**Strengths:**

The idea of applying spectral analysis to detect degenerative repetition in LLM outputs is novel. SpecRA offers a unique approach compared to existing penalty-based methods by focusing on post-generation detection through signal processing, providing a lightweight and non-intrusive solution.

**Weaknesses:**

1. Problem Importance: The problem of degenerative repetition is important, but the paper does not convincingly argue why it is a significant issue that warrants a completely new approach. Simple methods like repetition penalty or temperature adjustments can mitigate repetition without much loss in performance, and these are commonly used in practice. Also for long repetitive responses, techniques like early truncation could resolve these issues in a more cost-efficient manner, and the paper doesn’t explore these simpler alternatives in detail. While SpecRA is interesting, it’s unclear if the problem is as pressing as suggested.
2. Threshold Management: SpecRA requires setting a threshold for repetition detection, which can be cumbersome. The paper mentions that this threshold must be carefully adjusted for different vocabularies, which might require reconfiguration for each model. This adds a layer of complexity that could make the approach harder to deploy in production.

**Questions:**

Why is degenerative repetition such a critical issue that requires a new solution like SpecRA? Can existing solutions like repetition penalty or temperature adjustments not achieve similar results without introducing the complexity of SpecRA?

---

> ### Author Response · Authors · 2025-11-23
> **Response to Reviewer 8wU1 (1/2)**
>
> Dear Reviewer 8wU1,
>
> We appreciate your concerns about the problem importance and threshold management, and are happy to clarify our motivation and design choices.
>
> Regarding Weaknesses:
>
> > The problem of degenerative repetition is important, but the paper does not convincingly argue why it is a significant issue that warrants a completely new approach. Simple methods like repetition penalty or temperature adjustments can mitigate repetition without much loss in performance, and these are commonly used in practice.
>
> We agree that penalties and temperature adjustments are indeed widely used and effective in many scenarios, but they do not always suffice in industrial deployments. Developers have limited control if they rely on state-of-the-art proprietary models. For instance, GPT-5 <https://platform.openai.com/docs/guides/latest-model#gpt-5-1-parameter-compatibility> do not allow temperature adjustments, while Gemini <https://docs.cloud.google.com/vertex-ai/generative-ai/docs/models/gemini/2-5-pro> and Claude <https://platform.claude.com/docs/en/api/messages/create> do not allow custom penalties. As the industry adopts LLMs for a wider range of tasks, there is a growing demand for robust post-decoding solutions.
>
> While temperature adjustments and frequency/presence penalties can reduce repetition, they do so in a way that is difficult to control or reason about. In our experience, aggressively tuning these parameters may introduce undesirable side effects, such as more frequent code-switching or mixing in tokens from other languages during generation (e.g., a user requests Chinese output, but the model occasionally inserts Hindi or Russian words due to higher temperature). In contrast, SpecRA monitors the output stream without modifying the decoding process, so it does not compromise model performance or alter the distribution of generated content.
>
> > Also for long repetitive responses, techniques like early truncation could resolve these issues in a more cost-efficient manner, and the paper doesn’t explore these simpler alternatives in detail.
>
> This may stem from a misunderstanding of SpecRA's role. In practical deployments, a request to write single-page HTML may result in ~30,000 tokens of output and we cannot simply truncate by the length of the output. In order to early terminate the generation, we still need an reliable algorithm to tell us whether the LLM enters a loop, and that's what SpecRA is designed for. We agree that techniques like early truncation could resolve these issues in a more cost efficient manner, and that's exactly what we do on production deployments. Once repetition is detected by SpecRA, we early stop the generation and trigger resampling with higher temperature and extra prompts.
>
> > this threshold must be carefully adjusted for different vocabularies
>
> We thank the reviewer for raising the important topic of threshold management. This gives us a valuable opportunity to highlight a key strength of SpecRA's design.
>
> We would like to clarify on the threshold "must be carefully adjusted for different vocabularies.", on the contrary, a core advantage of SpecRA is that its threshold is robust and largely independent of the specific vocabulary, which can be calculated in a principled, data driven approach. To estimate the threshold, a practitioner can calculate the distribution of the repetition score on a relevant non-repetitive corpus, and fix a high quantile like 99.9th percentile as the threshold. This provides a clear, reproducible methodology. In the paper we are also providing recommended thresholds for code oriented tasks or general conversational workloads.

---

> ### Author Response · Authors · 2025-11-23
> **Response to Reviewer 8wU1 (2/2)**
>
> Regarding Questions:
>
> > Why is degenerative repetition such a critical issue that requires a new solution like SpecRA? Can existing solutions like repetition penalty or temperature adjustments not achieve similar results without introducing the complexity of SpecRA?
>
> Modifying the training or decoding process and tuning parameters is often challenging, and sometimes even infeasible, for most LLM application developers. In practice, slight changes to temperature or penalties often fail to eliminate degenerative repetition, while overly aggressive changes tend to introduce undesirable side effects and hurt performance. SpecRA is motivated by an industrial need for an explainable, post-decoding solution that suppresses repetition while preserving the quality of the generated content.
>
> Finally, although SpecRA introduces an additional component, the core implementation is lightweight and can be realized in just a few lines of code when using standard FFT libraries. We believe it's a reasonable trade-off for the benefits it brings. In a large scale, production like deployment, we observed a substantial reduction in repetition related issues after integrating the detector into agent workflows.
>
> Hope these clarifications have addressed your concerns, and we are glad to answer any further questions.
>
> Best regards,
>
> Authors of SpecRA

---

> > ### Comment · Reviewer_8wU1 · 2025-11-26
> >
> > Thank you for the detailed response.
> >
> > 1. Authors position SpecRA primarily for scenarios where model parameters (temperature, penalty) are inaccessible. However, it seems that the paper itself presents SpecRA as a general-purpose solution.
> >
> > 2. Maybe a comparison against straightforward baselines, such as n-gram self-overlap checks or other string-matching techniques, focusing on:
> > - Is SpecRA significantly faster or more accurate?
> > - Is the FFT-based periodicity analysis more interpretable for developers than simply identifying a repeating 20-gram?
> >
> > 3. "To estimate the threshold, a practitioner can calculate the distribution of the repetition score on a relevant non-repetitive corpus, and fix a high quantile like 99.9th percentile as the threshold."
> >
> > It seems that this process requires curating a calibration dataset and running an initial analysis, which is an additional step before applying.

---

> > > ### Author Response · Authors · 2025-11-26
> > > **Response to Reviewer 8wU1**
> > >
> > > Dear Reviewer 8wU1,
> > >
> > > Thank you for the insightful follow-up.
> > >
> > > While we highlighted the API scenario as it was the challenge that SpecRA originated from, we agree that SpecRA is a general-purpose solution. Even for model developers with full access, SpecRA offers specific advantages: it provides a quantitative metric for repetition reduction when tuning hyperparameters (temperature/penalty), acting as a reliable auxiliary evaluation framework and guiding more effective decoding strategies.
> > >
> > > Addressing your concern about baselines, we have added a comparison against k-mer(i.e. n-gram), SimHash, MinHash, and edit distance in both performance and robustness in Sec. 6, Figs 3-4 in the revised manuscript.
> > >
> > > In summary, SpecRA is faster than the above baselines we compared (~2.5x faster than k-mer/SimHash, ~25.9x faster than MinHash at N=4000) and tracks the edit distance accurately under noise. In contrast, our experiments show that compared to edit distance, k-mer(n-gram) methods tend to underestimate repetition when noise is present (as this creates new n-grams), while SimHash tends to overestimate it. MinHash is better calibrated but remains slower and less accurate than SpecRA.
> > >
> > > We use k=4(=n) for k-mer(n-gram) in our main experiments, as the result of k=20 would be worse: any minor change in the sequence would alter 20 distinct mers.
> > >
> > > The interpretability of SpecRA does not come primarily from the FFT implementation, but from the underlying quantity we compute: the autocorrelation based repetition score. Autocorrelation can be computed via convolution of the sequence with its time-reversed version, and FFT provides an efficient way to compute the convolution.
> > >
> > > The score $S_{\text{rep}}$ is an estimation of "the maximum fraction of tokens that repeat themselves after some lag $l$". For instance, the expectation of $S_{\text{rep}}$ for "ABCABD" is $\frac{2}{3}$, because 4 of 6 tokens repeat themselves after a lag of 3.
> > >
> > > Consider a model generating slightly varying code (treating each character as a token):
> > >
> > > ```
> > > def fun1():
> > > def fun2():
> > > def fun3():
> > > def fun4():
> > > def fun5():
> > > ```
> > >
> > > Here, the expected score $S_{\text{rep}}$ is $\frac{11}{12}$ (12 characters in each line, including `\n`), providing intuitive results for approximate repetition, whereas 4-grams will report half of the grams as unique.
> > >
> > > The raw output $S_{\text{rep}}$ represents how repetitive the text is. Threshold is required if we need a condition to determine whether approximate repetitions like "ABCABD" is repetitive or not, and this applies to all other baselines as well if we want a binary assertion.
> > >
> > > Without setting a threshold, SpecRA can still be applied. In practice, if no separate corpus is available, a practitioner can instead accumulate $S_{\text{rep}}$ statistics from their running application, thereby avoiding manual curation. Finally, for exact repetition, $S_{\text{rep}}=1$ is a definitive indicator, equivalent to exact string matching, requiring no threshold calibration.
> > >
> > > Best regards,
> > >
> > > Authors of SpecRA

---

### Official Review · Reviewer_vaVL · 2025-10-29

**Soundness:** 3
**Presentation:** 4
**Contribution:** 3
**Rating:** 4
**Confidence:** 3

**Summary:**

This paper introduces SpecRA, a new algorithm to detect degenerative repetition in LLM agents. It works by projecting the token stream onto a random complex-valued sequence and then using the FFT to efficiently compute its autocorrelation.

**Strengths:**

- The method's primary strengths are its efficiency, and its robustness to the approximate repetitions that break traditional exact-match detectors.
- The paper is very well motivated and is easy to read. The paper also openly mentions the limitations of the method, which is highly appreciated.

**Weaknesses:**

- In L195, the authors claim that the randomized projection into the complex plane makes the detector "robust to lexical variation" while preserving the overall periodic structure. Can the authors explain the connection between this projection and improved robustness to lexical variation?
- The decision to utilize only the real component of $R_l$ should be explicitly justified. While I could infer the motivation behind it, the exclusion of the imaginary component is not theoretically grounded in the text. A brief explanation would improve the conceptual clarity of the proposed method.
- The motivation behind the randomized projection step remains unclear. Mapping a vocabulary space of approximately 200K tokens into a 360$\degree$ complex plane appears arbitrary and potentially lossy. The authors mention this limitation briefly, but it would be more compelling to compare this approach with projection strategies, such as those based on embedding spaces or a higher-dimensional latent space.
- The experimental results are interesting but lack comparative context. Without evaluation against baseline models, for instance, n-gram based detectors or other lexical similarity measures, it is difficult to assess the effectiveness of SpecRA.

**Questions:**

All the questions and suggestions have been listed in the weaknesses.

While I do appreciate the authors addressing several limitations that I thought of while reading the paper (high FPR, failure in insertions / deletions & more), SpecRA, by itself, is a tool that would not be very useful realistically. Regardless, I would push this paper towards acceptance - based on the author's responses & for the theoretical insights and formulations provided by the authors.

---

> ### Author Response · Authors · 2025-11-23
> **Response to Reviewer vaVL (1/2)**
>
> Dear Reviewer vaVL,
>
> We appreciate your feedback and are glad to clarify your concerns and improve the manuscript.
>
> Regarding Weaknesses:
>
> > In L195, the authors claim that the randomized projection into the complex plane makes the detector "robust to lexical variation" while preserving the overall periodic structure. Can the authors explain the connection between this projection and improved robustness to lexical variation?
>
> The "design" in "This design makes the detector robust to lexical variations" refers to a two step mechanism: we first assign random phases to tokens, and then keep this map fixed for the entire run. The randomness ensures that non-repetitive text behaves like isotropic noise, while the fixed mapping guarantees that consistent lexical substitutions across periods preserve the same periodic structure in the complex signal.
>
> For instance, a sequence like (A B C A D C A B C) is mapped to $(e^{i\theta_A}, e^{i\theta_B}, e^{i\theta_C}, e^{i\theta_A}, e^{i\theta_D}, e^{i\theta_C}, e^{i\theta_A}, e^{i\theta_B}, e^{i\theta_C})$, thus preserving the overall periodic structure: A repeats itself every 3 tokens, and so does $e^{i\theta_A}$.
>
> > The decision to utilize only the real component of $R_l$ should be explicitly justified. While I could infer the motivation behind it, the exclusion of the imaginary component is not theoretically grounded in the text. A brief explanation would improve the conceptual clarity of the proposed method.
>
> We focus on the real part because for unit magnitude projections, $\Re(s_t s_{t-l}^*) = \cos(θ_t - θ_{t-l})$, which is exactly the cosine similarity between two unit vectors on the complex plane. This quantity already captures the degree of alignment: 1 for perfect matches, close to 0 for unrelated tokens. The imaginary part $\sin(θ_t - θ_{t-l})$ is the orthogonal component, which has zero mean and does not change the decision once the real part is fixed. Using $\Re(R_l)$ therefore provides a natural, interpretable similarity score and leads to a simpler concentration analysis, while $|R_l|$ does not materially change the detector's behavior but results in additional computational cost.
>
> We have added a brief explanation in the revised manuscript.
>
> > The motivation behind the randomized projection step remains unclear. Mapping a vocabulary space of approximately 200K tokens into a $360^\circ$ complex plane appears arbitrary and potentially lossy. The authors mention this limitation briefly, but it would be more compelling to compare this approach with projection strategies, such as those based on embedding spaces or a higher-dimensional latent space.
>
> Because of the "constant energy" constraint, all tokens should be mapped to a value with the same magnitude. Consider the case where projected tokens have different magnitudes: suppose we project A → 0.5, B → 0.2, C → 0.9 in real space $[0, 1]$, and compute the autocorrelation. A real valued version of SpecRA on "AABB" and "AACC" would then yield autocorrelation sequences [0.58, 0.49, 0.40, 0.49] ($S_{\text{rep}} = 0.84$) and [2.12, 1.96, 1.80, 1.96] ($S_{\text{rep}} = 0.92$), respectively. Under such a projection "AACC" appears more repetitive, not because its structure is inherently more repetitive, but because we are assigning unequal weights to different tokens. This naturally suggests projecting to at least 2D with equal norms.
>
> What about using the embeddings provided by models? First they do not necessarily possess unit magnitude so they treat different tokens unequally. Second, different words may have high cosine similarity, e.g. words from the same domain, see <https://arxiv.org/abs/1907.12009> and relevant topics on anisotropy of model embeddings. The anisotropy of embeddings can in principle increase the risk of false positives. Random projection into higher-dimensional hyperballs would have similar statistical properties but incurs higher computational and implementation cost without clear benefits for this application.
>
> And as a by-product of this deliberate design, we obtain a cleaner theoretical analysis by using complex numbers instead of 2D vectors.

---

> ### Author Response · Authors · 2025-11-23
> **Response to Reviewer vaVL (2/2)**
>
> > The experimental results are interesting but lack comparative context. Without evaluation against baseline models, for instance, n-gram based detectors or other lexical similarity measures, it is difficult to assess the effectiveness of SpecRA.
>
> Thank you for this insightful suggestion. To address this concern, we have added a systematic empirical comparison against edit distance, k-mer heuristics, SimHash and classical FFT-based matching (see Sec. 6, Figs. 3–4 in the revised manuscript).
>
> In a word, SpecRA closely tracks the edit distance baseline as noise increases, while SimHash/Classical FFT and K-mer systematically overestimate/underestimate the repetition of zipfian text. MinHash provides a relevantly calibrated estimate, while still not as accurate and fast as SpecRA.
>
> We measured the average runtime, and SpecRA is ~500x times faster(N=500) / ~300,000x faster(N=4000) than exhaustive edit distance search and remains competitive with lightweight k-mer heuristics. Due to space limits, we only summarize these findings here. Please refer to Sec. 6 and Appendix D for full details.
>
> ---
>
> And finally, we would like to clarify some possibly minor misunderstandings of SpecRA's limitations:
>
> - There are actually two types of "false positives": (i) true misdetections due to the randomness of the projection, which we did not observe in our experiments and they are controlled by our theoretical analysis, and (ii) "legitimate repetition" because the repetition is indeed the desired output of the user. For example, a user might be asking the model to translate a highly repetitive table to another language instead of modifying manually row by row, and in such a scenario, repetition is desired and should not be truncated. We are providing "legitimate repetition" as a warning to prompt practitioners to adjust their thresholds accordingly to their tasks. As noted in our paper, the "legitimate repetitions" we detected are with a threshold of 0.69, lower than the recommended threshold for coding tasks, while the taxonomy is built upon tasks across different domains from coding, data processing to document writing and translation. We have clarified this point in the revised manuscript.
>
> - In practice, models do not tend to produce repetition patterns that introduce indels in all repetitive blocks and in different positions. Instead, exact repetitions with minor substitutions, or minor insertions introduced by number increments are more common, and SpecRA is still effective in detecting them. While some research (e.g. <https://arxiv.org/pdf/2505.10402> Figure 1) reported more complicated repetitive patterns, they are not simple indels that a general purpose detector would be able to handle without knowledge of the structure of the task, and also beyond the "autorepetition" problem we are trying to solve. We have noted this in our limitations in the revised manuscript.
>
> Additional note from the authors: (i) we may relax the "constant energy" constraint and map specific tokens to zero, this allows us to ignore whitespaces and indentations. (ii) by adding zero valued padding tokens on each line to align phases, SpecRA can be extended to detect more complicated patterns of unequal line lengths, and (iii) with a different tokenizer from code to grammatical elements, SpecRA is able to analyze code at the grammatical level and become a code repetition detector.
>
> SpecRA was originally motivated by an industrial need for a robust autorepetition detector. In a large scale, production like deployment, we observed a substantial reduction in repetition related issues when integrating the detector into agent workflows.
>
> We hope these clarifications and revisions have fully addressed your concerns. We are grateful for your guidance in strengthening our paper and are happy to answer any further questions.
>
> Best regards,
>
> Authors of SpecRA

---

> > ### Comment · Reviewer_vaVL · 2025-11-24
> > **Response to Authors**
> >
> > I highly appreciate the author's rigorous response. Regardless, I have a few concerns:
> >
> > > "The randomness ensures that non-repetitive text behaves like isotropic noise, while the fixed mapping guarantees that consistent lexical substitutions"
> > - Don't continuous embedding spaces also have this property of fixed mappings? They are not invertible but - the point is - I wish to understand why the authors use a 360-degree complex plane. Perhaps an embedding space of higher / lower dimension with untrained (and normalized) weights could provide a better representation to the tokens. Untrained, in the context, refers to tokens being given random projections and no semantic information being captured.
> > - Given the computational efficiency problems (which is contradictory with the response provided to another weakness), have the authors explored how SpecRA holds up with a limited complex plane (for e.g. 270-degree or 180-degree)? I'd probably also want to see a larger space, given that SpecRA is 500x faster for N=500.
> > - A minor concern related to Reviewer `8wU1`'s comments: Out of curiosity, does SpecRA, in general, notice any decrement in detection numbers (not performance, but quantity) when we attempt to increase repetition penalty or decrease temperatures? This answer would be useful to model developers who can use an auxiliary detection framework to avoid repetition and make repetition penalty more effective.
> > - I highly appreciate the additional notes, those are excellent future directions and are worth exploring.

---

> > > ### Author Response · Authors · 2025-11-26
> > > **Response to Reviewer vaVL**
> > >
> > > Dear Reviewer vaVL,
> > >
> > > Thank you very much for your thoughtful comments and for your interest in SpecRA. We agree that, given SpecRA's significant speed advantage, computational cost should not prevent us from exploring higher dimensional spaces if they offer better representations.
> > >
> > > To address this, we implemented and evaluated a 128D variant, which projects tokens onto random unit vectors in a 64 dimensional complex space (normalized and untrained, on the surface of a 128D unit hypersphere), instead of the unit circle in the complex plane. We compared both the robustness (decay of repetition score under noise) and computational efficiency. The results are shown below:
> > >
> > > | Noise Level (W=1024, P=16) | SpecRA (complex) Score | SpecRA-128D (128D complex) Score |
> > > | :--- | :--- | :--- |
> > > | 0.00 | 1.000 | 1.000 |
> > > | 0.05 | 0.909 | 0.907 |
> > > | 0.10 | 0.821 | 0.818 |
> > > | 0.15 | 0.735 | 0.733 |
> > > | 0.20 | 0.655 | 0.653 |
> > > | 0.25 | 0.577 | 0.578 |
> > > | 0.30 | 0.507 | 0.507 |
> > >
> > > | Sequence Length (N) | SpecRA Time | SpecRA-128D (128D Complex) Time | Relative Slowdown |
> > > | :--- | :--- | :--- | :--- |
> > > | 500 | 0.041 ms | 0.293 ms | 7.1x |
> > > | 1000 | 0.068 ms | 0.686 ms | 10.1x |
> > > | 2000 | 0.132 ms | 1.689 ms | 12.8x |
> > > | 4000 | 0.266 ms | 3.875 ms | 14.5x |
> > >
> > > While it brings extra computational cost, the decay curves of the detection score under increasing noise are nearly identical.
> > >
> > > To answer "Why 2D is enough": The underlying value SpecRA computes is the autocorrelation of the sequence, every term $X_t \cdot X_{t+l}$ in the autocorrelation will be:
> > >
> > > - $1$, if $X_t = X_{t+l}$
> > > - $\cos(\theta)$, if $X_t \neq X_{t+l}$. Since the angle $\theta$ between $X_t$ and $X_{t+l}$ is uniform in $[0, 2\pi)$, the expectation $E[\cos(\theta)] = \frac{1}{2\pi}\int_0^{2\pi} \cos(\theta) d\theta = 0$ for any dimension $D \geq 2$.
> > >
> > > Thus we expect the sum of noises cancel out, leaving only the $1$ terms counting repetitions. As a result, the expectation of the autocorrelation is exactly the count of tokens that appears again after $l$ tokens.
> > >
> > > Moreover, because we are effectively computing cosine similarities between pairs of vectors, and any two vectors always lie in a 2D subspace, increasing the ambient dimension does not improve the signal‑to‑noise ratio beyond what can already be achieved in 2D.
> > >
> > > This zero-mean requirement also explains why we cannot use a limited complex plane. For example, on a 180-degree half circle, the angle of two different vectors $X, Y \sim \mathcal{U}[0, \pi]$ is a triangular distribution with expectation:
> > >
> > > $$
> > > E[\cos(X-Y)] = \frac{1}{\pi^2}\int_0^\pi\int_0^\pi \cos(x-y)dx dy = \frac{4}{\pi^2} \approx 0.405
> > > $$
> > >
> > > The noise terms no longer cancel out, introducing a systematic bias and breaking the detection mechanism.
> > >
> > > > Does SpecRA, in general, notice any decrement in detection numbers (not performance, but quantity) when we attempt to increase repetition penalty or decrease temperatures?
> > >
> > > This is an interesting question, and we employed an extra metric to observe the effect of temperature on detection number on live serving streams. We observed the quantity decreases from 0.28% to 0.12% when we increase the temperature from 0.6 to 1.0. So the answer is yes. (We interpret the "decrease temperature" as "increase temperature" in this context. Note that we focused on temperature adjustments as our model provider's API currently limits access to repetition penalty settings.)
> > >
> > > We fully agree with your perspective of using SpecRA as an auxiliary framework for designing better decoding strategies and guiding parameter tuning. This highlights a valuable practical application for model developers that goes beyond the scope of our initial submission.
> > >
> > > Thank you again for your insightful comments and for highlighting these important aspects of SpecRA.
> > >
> > > Best regards,
> > >
> > > Authors of SpecRA

---

### Official Review · Reviewer_UZ8b · 2025-10-31

**Soundness:** 4
**Presentation:** 3
**Contribution:** 3
**Rating:** 6
**Confidence:** 3

**Summary:**

This paper introduces SpecRA, a spectral-based algorithm for detecting degenerative repetition in LLM-based agents, where models become trapped in recursive loops generating near-identical sequences that cause task failure and computational waste. The method works by projecting each token to a unit-magnitude complex number via randomized phase assignment, then computing autocorrelation using FFT via the Wiener-Khinchin theorem to efficiently identify periodic patterns with O(W log W) complexity. The paper also provides theoretical guarantees including false-positive bounds and detection efficacy under $\epsilon$-mismatch conditions (Theorem 1), demonstrates robustness to substitutions, insertions, and deletions through synthetic experiments, and analyzes repetitive samples from >1M agent traces to build a taxonomy of four failure modes such as structural repetition and syntactic degradation. The authors recommend practical detection thresholds for different use cases.

**Strengths:**

+ Novel and theoretically grounded approach: The paper presents a creative solution by recasting text repetition detection as a signal processing problem. The use of randomized phase projection with FFT-based autocorrelation is elegant and well-motivated.

+ Strong theoretical foundation: Provides rigorous probabilistic guarantees through Lemma 1 (false-positive bounds using Hoeffding's inequality) and Theorem 1 (detection power under approximate periodicity), making the approach principled rather than heuristic.

+ Practical guidance: Provides actionable threshold recommendations based on empirical percentiles from diverse corpora, facilitating real-world adoption.

+ Clear presentation: The paper is well-written with good motivation, clear problem formulation, and effective visualizations (especially Figure 1).

+ Comprehensive empirical validation: The paper includes synthetic experiments (Section 6.1), real-world corpus analysis across multiple domains (Wikipedia, GitHub, agent traces), and builds a valuable taxonomy of repetition failure modes.

**Weaknesses:**

+ Missing baseline comparisons: The empirical analysis (Section 6) lacks comparisons with existing detection methods. Even if exact string matching and edit-distance methods have limitations, quantitative comparisons on the same test sets would strengthen claims about SpecRA's advantages in accuracy, speed, and robustness.


+ Taxonomy validation: The classification of 549 repetitive samples into four categories appears to be manual. The paper lacks details on: (a) inter-annotator agreement if multiple annotators were used, (b) whether the 264 excluded samples introduce selection bias, (c) validation that categories are mutually exclusive and comprehensive.

+ Real-world deployment details: The paper mentions "1.13M records of anonymized agent outputs" but provides minimal context about: the agents' tasks, which models were used, what triggered the repetitions, and how representative this dataset is.

**Questions:**

+ Baseline performance: Can you provide quantitative comparisons against n-gram overlap, suffix trees, or approximate string matching algorithms (e.g., using sliding windows with edit distance) on your test sets? Even if they're slower, understanding the accuracy trade-offs would be valuable.

+ False negative analysis: What is the false negative rate of SpecRA on your agent trace dataset? Are there patterns that SpecRA consistently misses, and what characterizes them?

+ Interaction with decoding parameters: How does SpecRA's detection rate vary with temperature, top-p, or other sampling parameters? Do higher temperatures reduce repetition occurrence or just change repetition patterns?

---

> ### Author Response · Authors · 2025-11-23
> **Response to Reviewer UZ8b (1/2)**
>
> Dear Reviewer UZ8b,
>
> We sincerely appreciate your thoughtful review and positive assessment of the paper's soundness, contribution, and theoretical foundations. We address your concerns below and have revised the manuscript accordingly.
>
> Regarding Weaknesses:
>
> **(1) Missing baseline comparisons**
>
> We thank you for highlighting the importance of such comparisons. We have added a systematic empirical comparison against edit distance, k-mer heuristics, SimHash and classical FFT-based matching (see Sec. 6, Figs. 3–4 in the revised manuscript).
>
> In summary, SpecRA closely tracks the edit distance baseline as noise increases, while SimHash/Classical FFT and K-mer systematically overestimate/underestimate the repetition of zipfian text. MinHash provides a relevantly calibrated estimate, while still not as accurate and fast as SpecRA.
>
> We measured the average runtime, and SpecRA is ~500x times faster(N=500) / ~300,000x faster(N=4000) than exhaustive edit distance search and remains competitive with lightweight k-mer heuristics.
>
> Due to space limits, we only summarize these findings here. Please refer to Sec. 6 and Appendix D for full details.
>
> **(2) Taxonomy validation**
>
> The 549 repetitive samples were annotated by multiple LLMs with major voting (as mentioned in "Use of LLMs" section), and then carefully verified and, when necessary, corrected by the authors, who take full responsibility for the final labels.
>
> The 264 examples were excluded because they were too short or incomplete to be classified reliably (e.g., due to network failure, user cancellation or safety filters). While they manifest repetitive features, it is uncertain whether the model would have entered an infinite loop (or equivalently whether the model will exit eventually after a some more periods). This may slightly shift the distribution, but we do not expect it to materially change the main qualitative trends reported in the paper
>
> During annotation, we allowed for an "Other" label, but found that most edge cases fit into "Structural repetition". Mutual exclusivity is ensured by a hierarchical decision process, which can be summarized as follows:
>
> 1. Does the pattern carry coherent meaning to a human reader?
>   - Yes → 2.
>   - No  → 3.
>   - Uncertain → (Other)
>
> 2. Given that the content is meaningful, is the repetition legitimate for the user's task (e.g., translating a repetitive table)?
>   - Yes → (Legitimate repetition)
>   - No  → (Structural repetition)
>   - Uncertain → (Other)
>
> 3. Given that the content is not meaningful to humans, does it still correspond to machine interpretable binary formats (e.g., base64 encoded multimedia content, binary file headers)?
>   - Yes → (Binary data generation)
>   - No  → (Syntactic degradation)
>   - Uncertain → (Other)
>
> We would like to also clarify that these categories are intended as a practically useful taxonomy rather than a strictly exhaustive ontology. We have incorporated a brief description of this annotation protocol and decision process into the revised manuscript for better transparency.
>
> **(3) Real-world deployment details**
>
> While we unfortunately cannot disclose specific deployment details due to NDA constraints and user privacy protection, we can provide aggregate context: the models used are state-of-the-art models (proprietary models without -mini/-flash suffix, or their equivalents with >100B parameters), and the tasks span the full software development lifecycle, data processing, analysis/visualization, and general NLP (document writing/translation, information processing, etc.) While most repetition events do not have a specific trigger, tasks involving code generation, data processing, and binary data manipulation are statistically more prone to repetition. We added a description in the revised manuscript to provide this aggregate deployment context while respecting NDA and privacy.

---

> ### Author Response · Authors · 2025-11-23
> **Response to Reviewer UZ8b (2/2)**
>
> Regarding Questions:
>
> **(1) Baseline performance**
>
> Please refer to our response in "Missing baseline comparisons" above.
>
> **(2) False negative analysis**
>
> Within the scope of the $(\varepsilon, p)$-approximate periodicity defined in Sec. 3, SpecRA has no observable false negatives on our dataset, consistent with our theoretical analysis. However, we are glad to discuss "semantic repetition" cases that might interest you: in coding scenarios, models may generate grammatically repetitive content where each block has different lengths (e.g., a series of similar imports but grow longer on each loop). SpecRA may not detect these if the phase alignment is disrupted by length variation. Detecting such cases would require abstracting tokens to grammatical elements, which is a specialized extension of SpecRA and beyond the scope of this paper. We have noted this in the revised manuscript.
>
> As an additional remark, (i) we may relax the "constant energy" constraint and map specific tokens to zero, this allows us to ignore whitespaces and indentations. (ii) by adding zero valued padding tokens on each line to align phases, SpecRA can be extended to detect more complicated patterns of unequal line lengths, and (iii) with a different tokenizer from code to grammatical elements, SpecRA is able to analyze code at the grammatical level and become a code repetition detector.
>
> **(3) Interaction with decoding parameters**
>
> SpecRA's detection efficacy is generally robust to decoding parameters. Higher temperatures do reduce the overall occurrence of repetition and tend to introduce more minor substitutions within repetitive loops. SpecRA is designed to handle these substitutions effectively (as shown in our noise robustness experiments).
>
> We are grateful for your detailed feedback and believe that the additional analyses and clarifications added in the revised manuscript will substantially strengthen the paper. Hope these clarifications have addressed your concerns, and we are glad to answer any further questions.
>
> Best regards,
>
> Authors of SpecRA

---

### Official Review · Reviewer_B63k · 2025-11-01

**Soundness:** 2
**Presentation:** 3
**Contribution:** 2
**Rating:** 4
**Confidence:** 3

**Summary:**

In this paper, the authors claim to introduce a new approach for detecting repetition degeneration in texts generated by LLMs, based on the discrete Fast Fourier Transform (FFT). The authors claim that the FFT is more scalable and robust for natural language repetitions than existing n-gram (k-mer)- and edit distance-based approaches. Authors provide proof that their method has a controllable false positive rate and exhibits exponentially improved performance with increasing attention frame size.In this papers the authors claim to introduce a new approach for the detection if repetition degeneration of texts generated by LLMs based on the discrete Fast Fourier Transform (FFT). The authors claim that FFT is more scalable and robust for natural language repetitions than the existing n-gram (k-mer) and edit distance- based approaches. Authors provide a proof that their method has a controllable false positive rate and exponentially improved performance with the increase in the attention frame size.

**Strengths:**

The detection of repetition degeneration in LLM output is a critical step in both its day-to-day usage, as highlighted by the authors, as well as in model training to suppress unwanted repetition at the Reinforcement Learning stage. The overall approach undertaken by the authors is sound and likely applicable in practice. Additionally, authors introduce an additional criterion for performance evaluation- "timely detection", which is relevant in the context of repetition detection but not a statistical criterion commonly used in other settings.

**Weaknesses:**

- While the overall citations are well-selected and consistent with the text, authors repeatedly cite (Holtzman et al., 2019) as the paper introducing or using the repetition penalty (eg, L040-047). However, this paper focused only on nucleus sampling, addressing the repetition text through a temperature-based sampling. It does not use repetition penalties.

- The authors do not sufficiently justify the need for their method. For instance, the k-mer sliding approaches that authors discard as insufficiently scalable (L091-097) can be applied to UTF-8 encodings of the character n-grams, effectively reducing the vocabulary back to 16 characters, making direct bioinformatics k-mer approaches computationally feasible again. Similarly, it is unclear how frequently the partial repetition problem is used to justify their approach.

- The authors do not seem to evaluate the computational performance of the proposed method, one of its advantages compared to existing methods, as claimed by the authors.

- Finally, the contribution of the paper is hard to identify and seems minor at first glance. Teglanceepetition analysis has been commonly cited as an application of the FFT to texts (e.g., https://math.stackexchange.com/questions/422948/fourier-transform-of-text; https://cp-algorithms.com/algebra/fft.html). The author's addition appears to be the random embedding of text in the complex plane before applying the FTT, but it is unclear why this step is important, except that it makes the proofs of Lemma 1 and Theorem 1 simpler. I am not sure, however, that such a contribution could justify acceptance to a conference of the notoriety of ICLR.


Minor comment: In Lemmas 1 and Theorem 1, tau is commonly used to denote time or position index in frequency analysis and integration; using it as a threshold may be confusing to readers from that background.

**Questions:**

- How frequent are the real-world scenarios in which the repetition degeneration occurs with minor variations, as suggested on L044-047? Most of the literature and reports on the topic suggest exact repetitions as the dominant degeneration mode.

- Is it possible to switch from a custom embedding of the tokens to the embedding provided by the model tokenizer? This is likely to lead to better performance by removing the need for a separate embedding step.

- Could you please clarify what you mean by "constant energy" on L049?

**Details Of Ethics Concerns:**

LLM usage: The Authors report extensive LLM usage, utilizing an LLM that they refuse to identify, for performing the original derivation of critical sections of the proof, which is a central contribution of the paper. While proof automation and assistance with LLMs are active areas of research, they are known to be somewhat haphazardous areas with extensive room for subtle errors that are hard to spot to non-expert human reviewers; and that often imitate existing proofs.

Based on a subtle citation mismatch with a high overall quality, I also suspect that the introduction and overview of existing methods were LLM-generated, as well as potentially the code implementation of FFT. It is unclear how it would connect with the commonly used LLM deployment Python code, although potentially doable within the stack of commercial LLM providers.

---

> ### Author Response · Authors · 2025-11-23
> **Response to Reviewer B63k (1/3)**
>
> Dear Reviewer B63k,
>
> We sincerely thank you for your careful and thorough review, and for identifying an important mistake that we and the other reviewers had missed.
>
> Regarding Weaknesses:
>
> **(1) citation mismatch**
>
> The citation in L40 was intended to be (OpenAI, 2025; ...), a reference to OpenAI's API documentation. OpenAI first introduced the frequency and presence penalties in their chat completions API. The three penalties (frequency, presence, and repetition) are now mainstream techniques to mitigate repetition and are supported by many LLM providers and inference frameworks. We have corrected this in the revised version.
>
> **(2) k-mer on UTF-8 encoding**
>
> We interpreted the suggestion of "reducing the vocabulary back to 16 characters" as referring to a hex encoding of UTF-8 code points like 0xFFFF, though a standard byte represents 256 values. UTF-8 is a variable-length encoding (1–4 bytes) designed for text encoding rather than analysis. Failing to respect character boundaries would disrupt the statistics in multilingual contexts. For instance, CJK unified ideographs are encoded as 3 bytes sharing a common prefix byte 0xE4–0xE9; a hex- or byte-wise k-mer approach may detect repetitions arising from the pattern of UTF-8 itself rather than from the underlying text. Moreover, LLM-generated Chinese text typically mixes 3-byte characters (Hanzi) with 1-byte punctuation (spaces, newlines, markdown syntax, etc.), making it difficult to choose a single k that preserves all character boundaries.
>
> However, we agree that our initial statement about k‑mer scalability was based on a rough upper-bound analysis rather than careful experiments. Based on the experiments below, we tested token level k-mer and found the complexity grows slower with the sequence length, as in real implementation we only need to store the unique k-mer presented. However, the performance degrades rapidly as noise increases, indicating its brittleness to noise. We have modified the manuscript to reflect this limitation.
>
> A practical UTF-8–based k-mer system would require a fairly complex strategy for choosing appropriate k(s) and a careful criterion for alerting repetition, making it less competitive. Moreover, SpecRA can detect repetition of length $L = W/2$ or even incomplete repetitions $L > W/2$, enabling more timely detection within the same window size.
>
> **(3) comparison with existing methods**
>
> To address this concern, we have added a systematic empirical comparison against edit distance, k-mer heuristics, SimHash, MinHash and classical FFT-based matching (see Sec. 6, Figs. 3–4 and Appendix D).
>
> In a word, SpecRA closely tracks the edit distance baseline as noise increases, while SimHash and K-mer systematically overestimate/underestimate the repetition of zipfian text sequence. Classical FFT-based approximate string matching by mapping $n$-th token to corresponding unit root $n \mapsto e^{2\pi i n/V}$ (<https://cp-algorithms.com/algebra/fft.html>) is also included in the experiments and it produces high S_rep scores due to the biased distribution of vocabulary.
>
> We measured the average runtime, and SpecRA is ~500x times faster(N=500) / ~300,000x faster(N=4000) than exhaustive edit distance search and remains competitive with lightweight k-mer heuristics.
>
> Due to space limits, we only summarize these findings here. Please refer to Sec. 6 for full details.
>
> **(4) contribution**
>
> We would like to clarify that the main novelty of SpecRA is not limited to using FFT per se, but in formulating and solving the non-intrusive streaming autorepetition detection problem: detecting degenerative repetition online from the output token stream alone without modifying the model or decoding, and with unknown pattern length and position. In the paper, SpecRA is defined by a set of design principles that together characterize a non-intrusive spectral detector: constant energy random projection of tokens into the complex plane to enforce isotropic embeddings, streaming autocorrelation over a sliding window, and provable control of false positives and misses under realistic vocabularies and noise.
>
> Within this framework, there is still room for alternative spectral instantiations (e.g., different random projection schemes or autocorrelation kernels) that satisfy the same non-intrusive autorepetition criteria. Our FFT-based design is the first such realization that is fully developed, analyzed and validated at scale, where we characterize score distributions, derive practical thresholds, and document a taxonomy of repetition modes. Exploring non-spectral detectors under the same non-intrusive problem setting is an interesting direction, but is outside the scope of the present paper.
>
> **(5) threshold notation**
>
> We have changed the notation of the threshold from $\tau$ to $S_{th}$ (and $\tau_{99.9}$ to $S_{99.9}$ respectively) to avoid confusion with the common use of $\tau$ as a time or position index.

---

> ### Author Response · Authors · 2025-11-23
> **Response to Reviewer B63k (2/3)**
>
> Regarding Questions:
>
> > How frequent are the real-world scenarios in which the repetition degeneration occurs with minor variations, as suggested on L044-047? Most of the literature and reports on the topic suggest exact repetitions as the dominant degeneration mode.
>
> Repetition degeneration with variations is, to our knowledge, still an emerging topic and has not yet been systematically characterized. LLM architectures and usage scenarios have evolved significantly since the early reports focusing on exact repetition, and LLMs are now deployed on a much broader range of tasks. Besides our own work, [Demystifying Repetition in LLM-based Code Generation](https://arxiv.org/abs/2504.12608) by Liu et al. (2025) proposes a taxonomy of 20 repetition patterns (Table IV), including many instances of what we define as approximate or structural repetition, such as similar functions and test cases with incrementing parameters (Figure 1). Furthermore, [Rethinking Repetition Problems of LLMs in Code Generation](https://arxiv.org/abs/2505.10402) by Dong et al. (2025) reports that exact repetition consists of only about $10%$ of repetition cases in their code-generation experiments (Figure 1 and Appendix A). These studies, together with our observations, suggest that non-exact forms of repetition are highly prevalent in modern LLMs.
>
> > Is it possible to switch from a custom embedding of the tokens to the embedding provided by the model tokenizer?
>
> It's technically possible to use the embeddings provided by the model tokenizer. However, this breaks the "constant energy" property that we rely on, as we explain below. Another notable reason is that the embeddings provided by models require high computational resources, and are not suitable for streaming applications.
>
> > Could you please clarify what you mean by "constant energy" on L049?
>
> Here, "energy" refers to the magnitude of the projected token and the "constant energy" condition means that all tokens are projected to vectors with the same magnitude. This is a key property of SpecRA. To see why, consider what happens if projected tokens have different magnitudes: suppose we project A → 0.5, B → 0.2, C → 0.9 in $[0, 1]$, and compute the autocorrelation. A real valued version of SpecRA on "AABB" and "AACC" would then yield autocorrelation sequences [0.58, 0.49, 0.40, 0.49] ($S_{\text{rep}} = 0.84$) and [2.12, 1.96, 1.80, 1.96] ($S_{\text{rep}} = 0.92$), respectively. Under such a projection "AACC" appears more repetitive, not because its structure is inherently more repetitive, but because we are assigning unequal weights to different tokens. This would force us to project tokens into at least two dimensions to restore a meaningful notion of repetition. By constraining all tokens to have the same magnitude, we treat them equally and obtain a cleaner theoretical analysis as a by-product.

---

> ### Author Response · Authors · 2025-11-23
> **Response to Reviewer B63k (3/3)**
>
> Regarding Ethics Concerns:
>
> We are grateful that you raised these concerns about our use of LLMs, and we take research integrity very seriously. However, we would like to clarify that the usage of LLMs in this work is strictly limited to paperwork. The idea, the proofs, the design and implementation, as well as the experiments were all developed by the authors. Below, we address the specific indicators that led to this misunderstanding.
>
> Like many human researchers, we create new terms on demand. In Sec. 1, we introduced the term "autorepetition" in the summarization of our contribution to distinguish the issue addressed by SpecRA from  "repetition analysis" you mentioned, and we explained the motivation above. It is highly improbable that an LLM would coin the specific term in this context.
>
> The original derivation of Lemma 1 and Theorem 1 was carried out by the authors, who take full responsibility for the validity of the proofs. To preserve the correctness of our original proofs, LLMs were only used to translate our derivations into initial LaTeX and to make symbol choices more conventional (this is what we meant by "drafting.") and the transcription are reviewed by the authors. The core of these results is the modeling of noise to an additive phase noise, and with a standard application of Hoeffding's inequality (which can be found in many machine learning textbooks), the exponential bound follows directly. We acknowledge that our original wording may have been imprecise that may lead readers to misinterpret the LLMs were being used to "perform the original derivation of critical sections of the proof" and have clarified this more explicitly in the revised version.
>
> We also thank you for pointing out the citation mismatch. Upon reviewing our source files, we suspect the mistake was likely caused by a wrong paste from BibTeX to LaTeX as the two entries are placed adjacently. Although the citation in Sec. 1 is incorrect, the intended reference can still be found from the related work at Sec. 2, where the three penalties are explicitly discussed together.
>
> The example implementation was originally written in Go because SpecRA was first developed for engineers and deployed in a Go-based LLM agent application. We recognize that Go may create a barrier for some readers, so in the revised version we have switched to a Python implementation to make our method more accessible to the broader community.
>
> The [Anonymous frontier model] mentioned in the Usage of LLMs section may reveal the authors' affiliation and violate the double-blind review process. We will disclose the model name in the final camera-ready version.
>
> We sincerely apologize for the lack of clarity that led to these valid concerns, and also appreciate your recognition of the overall quality of our work, which has encouraged us to further clarify the presentation and strengthen the technical exposition in the revised manuscript. We hope that this detailed explanation alleviates your worries about our research process, and have revised our manuscript accordingly for both clarity of the development of our work and the questions you raised.
>
> Best regards,
>
> Authors of SpecRA

---

> ### Author Response · Authors · 2025-12-03
> **Final Response to Reviewer B63k Regarding the Latest Revision**
>
> Dear Reviewer B63k,
>
> We realized why the original Go example may appear "AI generated". Code generated by LLMs often contains detailed comments, however many open source projects or utility functions in a well-maintained large codebase may also share this feature. While these comments may look verbose for readers with a signal processing background, they are necessary for engineers and readers without such background to better understand our method. Therefore we choose to retain these comments in the latest revised version.
>
> Your rigorous review also pushed us to re-examine our proofs. One potential concern we found is the independence of $Y_t$ and $Y_{t+l}$ in Lemma 1, since they share the random phase $\theta_t$. However, conditional on $\theta_t$, the term $Y_t = \cos(\theta_t - \theta_{t-l})$ depends only on $\theta_{t-l}$ and $Y_{t+l} = \cos(\theta_{t+l} - \theta_t)$ depends only on $\theta_{t+l}$. As $\theta_{t-l}$ and $\theta_{t+l}$ are independent and uniformly distributed, $Y_t$ and $Y_{t+l}$ are conditionally independent given $\theta_t$, and their conditional distributions do not depend on the value of $\theta_t$. This implies that $Y_t$ and $Y_{t+l}$ are indeed independent. By discarding a few wrap-around terms to avoid circular dependency, we obtain a collection of mutually independent bounded variables to which Hoeffding's inequality applies (or by other concentration inequalities we get a similar exponential bound with different constant/exponential factor). We have added a footnote in the revised paper to clarify this point.
>
> Regardless, we appreciate your detailed review, which helped us to improve the manuscript. We believe these clarifications and revisions address your concerns.
>
> Best regards,
>
> Authors of SpecRA

---

### Meta-Review · Area_Chair_T4ta · 2025-12-31

**Summary:**

This paper aims to mitigate the issue of “degenerative repetition” in LLM-based agents by introducing a method to detect self-repetitions in text. The proposed method is based on randomized projection and leverages FFT to compute the sequence’s autocorrelation.

**Reviewer Concerns:**

In general, the reviewers find the problem studied in this paper is interesting. The proposed method looks sound and promising. However, the reviewers also have concerns, including baseline comparison, experiments, the justification for design choices, and the writing (e.g., references). Hope the comments from the reviewers can improve this paper.

**Reviewer Scores:**

The reviewers share common concerns about this submission, and may maintain their scores due to these concerns.

---

### Decision · Program_Chairs · 2026-01-26

Reject